# Efficient Vector Representation for Documents through Corruption

**Minmin Chen**
Criteo Research
Palo Alto, CA 94301, USA
m.chen@criteo.com

## Abstract

We present an efficient document representation learning framework, Document Vector through Corruption (Doc2VecC). Doc2VecC represents each document as a simple average of word embeddings. It ensures a representation generated as such captures the semantic meanings of the document during learning. A corruption model is included, which introduces a data-dependent regularization that favors informative or rare words while forcing the embeddings of common and non-discriminative ones to be close to zero. Doc2VecC produces significantly better word embeddings than Word2Vec. We compare Doc2VecC with several state-of-the-art document representation learning algorithms. The simple model architecture introduced by Doc2VecC matches or out-performs the state-of-the-art in generating high-quality document representations for sentiment analysis, document classification as well as semantic relatedness tasks. The simplicity of the model enables training on billions of words per hour on a single machine. At the same time, the model is very efficient in generating representations of unseen documents at test time.

## 1 Introduction

Text understanding starts with the challenge of finding machine-understandable representation that captures the semantics of texts. Bag-of-words (BoW) and its N-gram extensions are arguably the most commonly used document representations. Despite its simplicity, BoW works surprisingly well for many tasks (Wang & Manning, 2012). However, by treating words and phrases as unique and discrete symbols, BoW often fails to capture the similarity between words or phrases and also suffers from sparsity and high dimensionality.

Recent works on using neural networks to learn distributed vector representations of words have gained great popularity. The well celebrated Word2Vec (Mikolov et al., 2013a), by learning to predict the target word using its neighboring words, maps words of similar meanings to nearby points in the continuous vector space. The surprisingly simple model has succeeded in generating high-quality word embeddings for tasks such as language modeling, text understanding and machine translation. Word2Vec naturally scales to large datasets thanks to its simple model architecture. It can be trained on billions of words per hour on a single machine.

Paragraph Vectors (Le & Mikolov, 2014) generalize the idea to learn vector representation for documents. A target word is predicted by the word embeddings of its neighbors in together with a unique document vector learned for each document. It outperforms established document representations, such as BoW and Latent Dirichlet Allocation (Blei et al., 2003), on various text understanding tasks (Dai et al., 2015). However, two caveats come with this approach: 1) the number of parameters grows with the size of the training corpus, which can easily go to billions; and 2) it is expensive to generate vector representations for unseen documents at test time.

We propose an efficient model architecture, referred to as Document Vector through Corruption (Doc2VecC), to learn vector representations for documents. It is motivated by the observation that linear operations on the word embeddings learned by Word2Vec can sustain substantial amount of syntactic and semantic meanings of a phrase or a sentence (Mikolov et al., 2013b). For example, vec("Russia") + vec("river") is close to vec("Volga River") (Mikolov & Dean, 2013), and

vec("king") - vec("man") + vec("women") is close to vec("queen") (Mikolov et al., 2013b). In Doc2VecC, we represent each document as a simple average of the word embeddings of all the words in the document. In contrast to existing approaches which post-process learned word embeddings to form document representation (Socher et al., 2013; Mesnil et al., 2014), Doc2VecC enforces a meaningful document representation can be formed by averaging the word embeddings **during learning**. Furthermore, we include a corruption model that randomly remove words from a document during learning, a mechanism that is critical to the performance and learning speed of our algorithm.

Doc2VecC has several desirable properties: 1. The model complexity of Doc2VecC is decoupled from the size of the training corpus, depending only on the size of the vocabulary; 2. The model architecture of Doc2VecC resembles that of Word2Vec, and can be trained very efficiently; 3. The new framework implicitly introduces a data-dependent regularization, which favors rare or informative words and suppresses words that are common but not discriminative; 4. Vector representation of a document can be generated by simply averaging the learned word embeddings of all the words in the document, which significantly boost test efficiency; 5. The vector representation generated by Doc2VecC matches or beats the state-of-the-art for sentiment analysis, document classification as well as semantic relatedness tasks.

## 2 RELATED WORKS AND NOTATIONS

Text representation learning has been extensively studied. Popular representations range from the simplest BoW and its term-frequency based variants (Salton & Buckley, 1988), language model based methods (Croft & Lafferty, 2013; Mikolov et al., 2010; Kim et al., 2015), topic models (Deerwester et al., 1990; Blei et al., 2003), Denoising Autoencoders and its variants (Vincent et al., 2008; Chen et al., 2012), and distributed vector representations (Mesnil et al., 2014; Le & Mikolov, 2014; Kiros et al., 2015). Another prominent line of work includes learning task-specific document representation with deep neural networks, such as CNN (Zhang & LeCun, 2015) or LSTM based approaches (Tai et al., 2015; Dai & Le, 2015).

In this section, we briefly introduce Word2Vec and Paragraph Vectors, the two approaches that are most similar to ours. There are two well-know model architectures used for both methods, referred to as Continuous Bag-of-Words (CBoW) and Skipgram models (Mikolov et al., 2013a). In this work, we focus on CBoW. Extending to Skipgram is straightforward. Here are the notations we are going to use throughout the paper:

$\mathcal{D} = \{D_1, \cdots, D_n\}$: a training corpus of size $n$, in which each document $D_i$ contains a variable-length sequence of words $w_i^1, \cdots, w_i^{T_i}$;

$V$: the vocabulary used in the training corpus, of sizes $v$;

$\mathbf{x} \in \mathcal{R}^{v \times 1}$: BoW of a document, where $x_j = 1$ iff word $j$ does appear in the document.

$\mathbf{c}^t \in \mathcal{R}^{v \times 1}$: BoW of the local context $w^{t-k}, \cdots, w^{t-1}, w^{t+1}, \cdots, w^{t+k}$ at the target position $t$. $c_j^t = 1$ iff word $j$ appears within the sliding window of the target;

$\mathbf{U} \in \mathcal{R}^{h \times v}$: the projection matrix from the input space to a hidden space of size $h$. We use $\mathbf{u}_w$ to denote the column in $\mathbf{U}$ for word $w$, i.e., the "input" vector of word $w$;

$\mathbf{V}^\top \in \mathcal{R}^{v \times h}$: the projection matrix from the hidden space to output. Similarly, we use $\mathbf{v}_w$ to denote the column in $\mathbf{V}$ for word $w$, i.e., the "output" vector of word $w$.

**Word2Vec.** Word2Vec proposed a neural network architecture of an input layer, a projection layer parameterized by the matrix $\mathbf{U}$ and an output layer by $\mathbf{V}^\top$. It defines the probability of observing the target word $w^t$ in a document $D$ given its local context $\mathbf{c}^t$ as

$$P(w^t|\mathbf{c}^t) = \frac{\exp(\mathbf{v}_{w^t}^\top \mathbf{U}\mathbf{c}^t)}{\sum_{w' \in V} \exp(\mathbf{v}_{w'}^\top \mathbf{U}\mathbf{c}^t)}$$

The word vectors are then learned to maximize the log likelihood of observing the target word at each position of the document. Various techniques (Mitchell & Lapata, 2010; Zanzotto et al., 2010; Yessenalina & Cardie, 2011; Grefenstette et al., 2013; Socher et al., 2013; Kusner et al., 2015)

have been studied to generate vector representations of documents from word embeddings, among which the simplest approach is to use weighted average of word embeddings. Similarly, our method forms document representation by averaging word embeddings of all the words in the document. Differently, as our model encodes the compositionality of words in the learned word embeddings, heuristic weighting at test time is not required.

**Paragraph Vectors.** Paragraph Vectors, on the other hands, explicitly learns a document vector with the word embeddings. It introduces another projection matrix $\mathbf{D} \in \mathcal{R}^{h \times n}$. Each column of $\mathbf{D}$ acts as a memory of the global topic of the corresponding document. It then defines the probability of observing the target word $w^t$ in a document $D$ given its local context $\mathbf{c}^t$ as

$$P(w^t | \mathbf{c}^t, \mathbf{d}) = \frac{\exp(\mathbf{v}_{w^t}^\top (\mathbf{U}\mathbf{c}^t + \mathbf{d}))}{\sum_{w' \in V} \exp(\mathbf{v}_{w'}^\top (\mathbf{U}\mathbf{c}^t + \mathbf{d}))}$$

where $\mathbf{d} \in \mathbf{D}$ is the vector representation of the document. As we can see from this formula, the complexity of Paragraph Vectors grows with not only the size of the vocabulary, but also the size of the training corpus. While we can reasonably limit the size of a vocabulary to be within a million for most datasets, the size of a training corpus can easily go to billions. What is more concerning is that, in order to come up with the vector representations of unseen documents, we need to perform an expensive inference by appending more columns to $\mathbf{D}$ and gradient descent on $\mathbf{D}$ while fixing other parameters of the learned model.

## 3 METHOD

Several works (Mikolov & Dean, 2013; Mikolov et al., 2013b) showcased that syntactic and semantic regularities of phrases and sentences are reasonably well preserved by adding or subtracting word embeddings learned through Word2Vec. It prompts us to explore the option of simply representing a document as an average of word embeddings. Figure 1 illustrates the new model architecture.

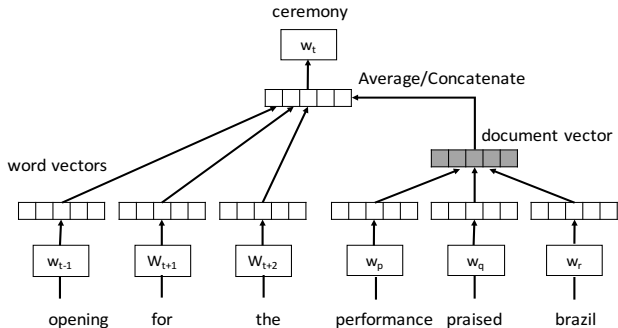

Figure 1: A new framework for learning document vectors.

Similar to Word2Vec or Paragraph Vectors, Doc2VecC consists of an input layer, a projection layer as well as an output layer to predict the target word, "ceremony" in this example. The embeddings of neighboring words ("opening", "for", "the") provide local context while the vector representation of the entire document (shown in grey) serves as the global context. In contrast to Paragraph Vectors, which directly learns a unique vector for each document, Doc2VecC represents each document as an average of the embeddings of words randomly sampled from the document ("performance" at position $p$, "praised" at position $q$, and "brazil" at position $r$).

Huang et al. (2012) also proposed the idea of using average of word embeddings to represent the global context of a document. Different from their work, we choose to corrupt the original document by randomly removing significant portion of words, and represent the document using only the embeddings of the words remained. This corruption mechanism offers us great speedup during training as it significantly reduces the number of parameters to update in back propagation. At the same time, as we are going to detail in the next section, it introduces a special form of regularization, which brings great performance improvement.

Here we describe the stochastic process we used to generate a global context at each update. The global context, which we denote as $\tilde{\mathbf{x}}$, is generated through a unbiased *mask-out/drop-out* corruption, in which we randomly overwrites each dimension of the original document $\mathbf{x}$ with probability $q$. To make the corruption unbiased, we set the uncorrupted dimensions to $1/(1-q)$ times its original value. Formally,

$$\tilde{x}_d = \begin{cases} 0, & \text{with probability } q \\ \frac{x_d}{1-q}, & \text{otherwise} \end{cases} \tag{1}$$

Doc2VecC then defines the probability of observing a target word $w^t$ given its local context $\mathbf{c}^t$ as well as the global context $\tilde{\mathbf{x}}$ as

$$P(w^t|\mathbf{c}^t, \tilde{\mathbf{x}}) = \frac{\exp(\mathbf{v}_{w^t}^\top(\overbrace{\mathbf{Uc}^t}^{\text{local context}} + \overbrace{\frac{1}{T}\mathbf{U}\tilde{\mathbf{x}}}^{\text{global context}}))}{\sum_{w' \in V} \exp(\mathbf{v}_{w'}^\top(\mathbf{Uc}^t + \frac{1}{T}\mathbf{U}\tilde{\mathbf{x}}))} \tag{2}$$

Here $T$ is the length of the document. Exactly computing the probability is impractical, instead we approximate it with negative sampling (Mikolov et al., 2013a).

$$\begin{aligned} f(w, \mathbf{c}, \tilde{\mathbf{x}}) &\equiv \log P(w^t|\mathbf{c}^t, \tilde{\mathbf{x}}) \\ &\approx \log \sigma\left(\mathbf{v}_w^\top(\mathbf{Uc} + \frac{1}{T}\mathbf{U}\tilde{\mathbf{x}})\right) + \sum_{w' \sim P_v} \log \sigma\left(-\mathbf{v}_{w'}^\top(\mathbf{Uc} + \frac{1}{T}\mathbf{U}\tilde{\mathbf{x}})\right) \end{aligned} \tag{3}$$

here $P_v$ stands for a uniform distribution over the terms in the vocabulary. The two projection matrices $\mathbf{U}$ and $\mathbf{V}$ are then learned to minimize the loss:

$$\ell = -\sum_{i=1}^{n} \sum_{t=1}^{T_i} f(w_i^t, \mathbf{c}_i^t, \tilde{\mathbf{x}}_i^t) \tag{4}$$

Given the learned projection matrix $\mathbf{U}$, we then represent each document simply as an average of the embeddings of the words in the document,

$$\mathbf{d} = \frac{1}{T} \sum_{w \in D} \mathbf{u}_w. \tag{5}$$

We are going to elaborate next why we choose to corrupt the original document with the corruption model in eq.(1) during learning, and how it enables us to simply use the average word embeddings as the vector representation for documents at test time.

## 3.1 CORRUPTION AS DATA-DEPENDENT REGULARIZATION

We approximate the log likelihood for each instance $f(w, \mathbf{c}, \tilde{\mathbf{x}})$ in eq.(4) with its Taylor expansion with respect to $\tilde{\mathbf{x}}$ up to the second-order (Van Der Maaten et al., 2013; Wager et al., 2013; Chen et al., 2014). Concretely, we choose to expand at the mean of the corruption $\mu_{\mathbf{x}} = \mathbb{E}_{p(\tilde{\mathbf{x}}|\mathbf{x})}[\tilde{\mathbf{x}}]$:

$$f(w, \mathbf{c}, \tilde{\mathbf{x}}) \approx f(w, \mathbf{c}, \mu_{\mathbf{x}}) + (\tilde{\mathbf{x}} - \mu_{\mathbf{x}})^\top \nabla_{\tilde{\mathbf{x}}} f + \frac{1}{2}(\tilde{\mathbf{x}} - \mu_{\mathbf{x}})^\top \nabla_{\tilde{\mathbf{x}}}^2 f (\tilde{\mathbf{x}} - \mu_{\mathbf{x}})$$

where $\nabla_{\tilde{\mathbf{x}}} f$ and $\nabla_{\tilde{\mathbf{x}}}^2 f$ are the first-order (i.e., gradient) and second-order (i.e., Hessian) of the log likelihood with respect to $\tilde{\mathbf{x}}$. Expansion at the mean $\mu_{\mathbf{x}}$ is crucial as shown in the following steps. Let us assume that for each instance, we are going to sample the global context $\tilde{\mathbf{x}}$ infinitely many times, and thus compute the expected log likelihood with respect to the corrupted $\tilde{\mathbf{x}}$.

$$\mathbb{E}_{p(\tilde{\mathbf{x}}|\mathbf{x})}[f(w, \mathbf{c}, \tilde{\mathbf{x}})] \approx f(w, \mathbf{c}, \mu_{\mathbf{x}}) + \frac{1}{2}\text{tr}\left(\mathbb{E}[(\tilde{\mathbf{x}} - \mathbf{x})(\tilde{\mathbf{x}} - \mathbf{x})^\top]\nabla_{\tilde{\mathbf{x}}}^2 f\right)$$

The linear term disappears as $\mathbb{E}_{p(\tilde{\mathbf{x}}|\mathbf{x})}[\tilde{\mathbf{x}} - \mu_{\mathbf{x}}] = 0$. We substitute in $\mathbf{x}$ for the mean $\mu_{\mathbf{x}}$ of the corrupting distribution (unbiased corruption) and the matrix $\Sigma_{\mathbf{x}} = \mathbb{E}[(\tilde{\mathbf{x}} - \mu_{\mathbf{x}})(\tilde{\mathbf{x}} - \mu_{\mathbf{x}})^\top]$ for the variance, and obtain

$$\mathbb{E}_{p(\tilde{\mathbf{x}}|\mathbf{x})}[f(w, \mathbf{c}, \tilde{\mathbf{x}})] \approx f(w, \mathbf{c}, \mathbf{x}) + \frac{1}{2}\text{tr}\left(\Sigma_{\mathbf{x}}\nabla_{\tilde{\mathbf{x}}}^2 f\right) \tag{6}$$

As each word in a document is corrupted independently of others, the variance matrix $\Sigma_\mathbf{x}$ is simplified to a diagonal matrix with $j^{th}$ element equals $\frac{q}{1-q}x_j^2$. As a result, we only need to compute the diagonal terms of the Hessian matrix $\nabla_\mathbf{\tilde{x}}^2 f$.

The $j^{th}$ dimension of the Hessian's diagonal evaluated at the mean $\mathbf{x}$ is given by

$$\frac{\partial^2 f}{\partial x_j^2} = -\sigma_{w,\mathbf{c},\mathbf{x}}(1 - \sigma_{w,\mathbf{c},\mathbf{x}})(\frac{1}{T}\mathbf{v}_w^\top \mathbf{u}_j)^2 - \sum_{w' \sim P_v} \sigma_{w',\mathbf{c},\mathbf{x}}(1 - \sigma_{w',\mathbf{c},\mathbf{x}})(\frac{1}{T}\mathbf{v}_{w'}^\top \mathbf{u}_j)^2$$

Plug the Hessian matrix and the variance matrix back into eq.(6), and then back to the loss defined in eq.(4), we can see that Doc2VecC intrinsically minimizes

$$\ell = -\sum_{i=1}^{n}\sum_{t=1}^{T_i} f(w_i^t, \mathbf{c}_i^t, \mathbf{x}_i) + \frac{q}{1-q}\sum_{j=1}^{v} R(\mathbf{u}_j) \qquad (7)$$

Each $f(w_i^t, \mathbf{c}_i^t, \mathbf{x}_i)$ in the first term measures the log likelihood of observing the target word $w_i^t$ given its local context $\mathbf{c}_i^t$ and the document vector $\mathbf{d}_i = \frac{1}{T}\mathbf{U}\mathbf{x}_i$. *As such, Doc2VecC enforces that a document vector generated by averaging word embeddings can capture the global semantics of the document, and fill in information missed in the local context.*

The second term here is a data-dependent regularization. The regularization on the embedding $\mathbf{u}_j$ of each word $j$ takes the following form,

$$R(\mathbf{u}_j) \propto \sum_{i=1}^{n}\sum_{t=1}^{T_i} x_{ij}^2 \left[ \sigma_{w_i^t,\mathbf{c}_i^t,\mathbf{x}_i}(1 - \sigma_{w_i^t,\mathbf{c}_i^t,\mathbf{x}_i})(\frac{1}{T}\mathbf{v}_{w_i^t}^\top \mathbf{u}_j)^2 + \sum_{w' \sim P_v} \sigma_{w',\mathbf{c}_i^t,\mathbf{x}_i}(1 - \sigma_{w',\mathbf{c}_i^t,\mathbf{x}_i})(\frac{1}{T}\mathbf{v}_{w'}^\top \mathbf{u}_j)^2 \right]$$

where $\sigma_{w,\mathbf{c},\mathbf{x}} = \sigma(\mathbf{v}_w^\top(\mathbf{U}\mathbf{c} + \frac{1}{T}\mathbf{U}\mathbf{x}))$ prescribes the confidence of predicting the target word $w$ given its neighboring context $\mathbf{c}$ as well as the document vector $\mathbf{d} = \frac{1}{T}\mathbf{U}\mathbf{x}$.

Closely examining $R(\mathbf{u}_j)$ leads to several interesting findings: 1. the regularizer penalizes more on the embeddings of common words. A word $j$ that frequently appears across the training corpus, i.e, $x_{ij} = 1$ often, will have a bigger regularization than a rare word; 2. on the other hand, the regularization is modulated by $\sigma_{w,\mathbf{c},\mathbf{x}}(1 - \sigma_{w,\mathbf{c},\mathbf{x}})$, which is small if $\sigma_{w,\mathbf{c},\mathbf{x}} \to 1$ or 0. In other words, if $\mathbf{u}_j$ is critical to a confident prediction $\sigma_{w,\mathbf{c},\mathbf{x}}$ when it is active, then the regularization is diminished. Similar effect was observed for dropout training for logistic regression model (Wager et al., 2013) and denoising autoencoders (Chen et al., 2014).

## 4 EXPERIMENTS

We evaluate Doc2VecC on a sentiment analysis task, a document classification task and a semantic relatedness task, along with several document representation learning algorithms. All experiments can be reproduced using the code available at https://github.com/mchen24/iclr2017

### 4.1 BASELINES

We compare against the following document representation baselines: **bag-of-words (BoW)**; **Denoising Autoencoders (DEA) (Vincent et al., 2008)**, a representation learned from reconstructing original document $\mathbf{x}$ using corrupted one $\mathbf{\tilde{x}}$. SDAs have been shown to be the state-of-the-art for sentiment analysis tasks (Glorot et al., 2011). We used Kullback-Liebler divergence as the reconstruction error and an affine encoder. To scale up the algorithm to large vocabulary, we only take into account the non-zero elements of $\mathbf{x}$ in the reconstruction error and employed negative sampling for the remainings; **Word2Vec (Mikolov et al., 2013a)+IDF**, a representation generated through weighted average of word vectors learned using Word2Vec; **Doc2Vec (Le & Mikolov, 2014)**; **Skip-thought Vectors(Kiros et al., 2015)**, a generic, distributed sentence encoder that extends the Word2Vec skip-gram model to sentence level. It has been shown to produce highly generic sentence representations that apply to various natural language processing tasks. We also include **RNNLM (Mikolov et al., 2010)**, a recurrent neural network based language model in the comparison. In the semantic relatedness task, we further compare to **LSTM-based methods** (Tai et al., 2015) that have been reported on this dataset.

Table 1: Classification error of a linear classifier trained on various document representations on the Imdb dataset.

| Model | Error rate % (include test) | Error rate % (exclude test) |
|---|---|---|
| Bag-of-Words (BOW) | 12.53 | 12.59 |
| RNN-LM | 13.59 | 13.59 |
| Denoising Autoencoders (DEA) | 11.58 | 12.54 |
| Word2Vec + AVG | 12.11 | 12.69 |
| Word2Vec + IDF | 11.28 | 11.92 |
| Paragraph Vectors | 10.81 | 12.10 |
| Skip-thought Vectors | - | 17.42 |
| Doc2VecC | **10.48** | **11.70** |

## 4.2 SENTIMENT ANALYSIS

For sentiment analysis, we use the IMDB movie review dataset. It contains 100,000 movies reviews categorized as either positive or negative. It comes with predefined train/test split (Maas et al., 2011): 25,000 reviews are used for training, 25,000 for testing, and the rest as unlabeled data. The two classes are balanced in the training and testing sets. We remove words that appear less than 10 times in the training set, resulting in a vocabulary of 43,375 distinct words and symbols.

**Setup.** We test the various representation learning algorithms under two settings: one follows the same protocol proposed in (Mesnil et al., 2014), where representation is learned using all the available data, including the test set; another one where the representation is learned using training and unlabeled set only. For both settings, a linear support vector machine (SVM) (Fan et al., 2008) is trained afterwards on the learned representation for classification. For Skip-thought Vectors, we used the generic model[1] trained on a much bigger book corpus to encode the documents. A vector of 4800 dimensions, first 2400 from the uni-skip model, and the last 2400 from the bi-skip model, are generated for each document. In comparison, all the other algorithms produce a vector representation of size 100. The supervised RNN-LM is learned on the training set only. The hyper-parameters are tuned on a validation set subsampled from the training set.

**Accuracy.** Comparing the two columns in Table 1, we can see that all the representation learning algorithms benefits from including the testing data during the representation learning phrase. Doc2VecC achieved similar or even better performance than Paragraph Vectors. Both methods outperforms the other baselines, beating the BOW representation by 15%. In comparison with Word2Vec+IDF, which applies post-processing on learned word embeddings to form document representation, Doc2VecC naturally enforces document semantics to be captured by averaged word embeddings during training. This leads to better performance. Doc2VecC reduces to Denoising Autoencoders (DEA) if the local context words are removed from the paradigm shown in Figure 1. By including the context words, Doc2VecC allows the document vector to focus more on capturing the global context. Skip-thought vectors perform surprisingly poor on this dataset comparing to other methods. We hypothesized that it is due to the length of paragraphs in this dataset. The average length of paragraphs in the IMDB movie review dataset is 296.5, much longer than the ones used for training and testing in the original paper, which is in the order of 10. As noted in (Tai et al., 2015), the performance of LSTM based method (similarly, the gated RNN used in Skip-thought vectors) drops significantly with increasing paragraph length, as it is hard to preserve state over long sequences of words.

**Time.** Table 2 summarizes the time required by these algorithms to learn and generate the document representation. Word2Vec is the fastest one to train. Denoising Autoencoders and Doc2VecC second that. The number of parameters that needs to be back-propagated in each update was increased by the number of surviving words in $\tilde{x}$. We found that both models are not sensitive to the corruption rate $q$ in the noise model. Since the learning time decreases with higher corruption rate, we used $q = 0.9$ throughout the experiments. Paragraph Vectors takes longer time to train as there are more parameters (linear to the number of document in the learning set) to learn. At test time, Word2Vec+IDF, DEA and Doc2VecC all use (weighted) averaging of word embeddings as document

---

[1]available at https://github.com/ryankiros/skip-thoughts

Table 2: Learning time and representation generation time required by different representation learning algorithms.

| Model | Learning time | Generation time |
|---|---|---|
| Denoising Autoencoders | 3m 23s | 7s |
| Word2Vec + IDF | 2m 33s | 7s |
| Paragraph Vectors | 4m 54s | 4m 17s |
| Skip-thought | 2h | 2h |
| Doc2VecC | 4m 30s | 7s |

Table 3: Words with embeddings closest to 0 learned by different algorithms.

| | |
|---|---|
| Word2Vec | harp(118) distasteful(115) switzerland(101) shabby(103) fireworks(101) heavens(100) thornton(108) endeavor(100) dense(108) circumstance(119) debacle(103) |
| ParaVectors | harp(118) dense(108) reels(115) fireworks(101) its'(103) unnoticed(112) pony(102) fulfilled(107) heavens(100) bliss(110) canned(114) shabby(103) debacle(103) |
| Doc2VecC | ,(1099319) .(1306691) the(1340408) of(581667) and(651119) up(49871) to(537570) that(275240) time(48205) endeavor(100) here(21118) way(31302) own(13456) |

representation. Paragraph Vectors, on the other hand, requires another round of inference to produce the vector representation of unseen test documents. It takes Paragraph Vectors 4 minutes and 17 seconds to infer the vector representations for the 25,000 test documents, in comparison to 7 seconds for the other methods. As we did not re-train the Skip-thought vector models on this dataset, the training time[2] reported in the table is the time it takes to generate the embeddings for the 25,000 training documents. Due to repeated high-dimensional matrix operations required for encoding long paragraphs, it takes fairly long time to generate the representations for these documents. Similarly for testing. The experiments were conducted on a desktop with Intel i7 2.2Ghz cpu.

**Data dependent regularization.** As explained in Section 3.1, the corruption introduced in Doc2VecC acts as a data-dependent regularization that suppresses the embeddings of frequent but uninformative words. Here we conduct an experiment to exam the effect. We used a cutoff of 100 in this experiment. Table 3 lists the words having the smallest $l_2$ norm of embeddings found by different algorithms. The number inside the parenthesis after each word is the number of times this word appears in the learning set. In word2Vec or Paragraph Vectors, the least frequent words have embeddings that are close to zero, despite some of them being indicative of sentiment such as debacle, bliss and shabby. In contrast, Doc2VecC manages to clamp down the representation of words frequently appear in the training set, but are uninformative, such as symbols and stop words.

**Subsampling frequent words.** Note that for all the numbers reported, we applied the trick of subsampling of frequent words introduced in (Mikolov & Dean, 2013) to counter the imbalance between frequent and rare words. It is critical to the performance of simple Word2Vec+AVG as the sole remedy to diminish the contribution of common words in the final document representation. If we were to remove this step, the error rate of Word2Vec+AVG will increases from 12.1% to 13.2%. Doc2VecC on the other hand naturally exerts a stronger regularization toward embeddings of words that are frequent but uninformative, therefore does not rely on this trick.

## 4.3 WORD ANALOGY

In table 3, we demonstrated that the corruption model introduced in Doc2VecC dampens the embeddings of words which are common and non-discriminative (stop words). In this experiment, we are going to quantatively compare the word embeddings generated by Doc2VecC to the ones generated by Word2Vec, or Paragraph Vectors on the word analogy task introduced by Mikolov et al. (2013a). The dataset contains five types of semantic questions, and nine types of syntactic questions, with a total of 8,869 semantic and 10,675 syntactic questions. The questions are answered through simple linear algebraic operations on the word embeddings generated by different methods. Please refer to the original paper for more details on the evaluation protocol.

---

[2]As reported in the original paper, training of the skip-thought vector model on the book corpus dataset takes around 2 weeks on GPU.

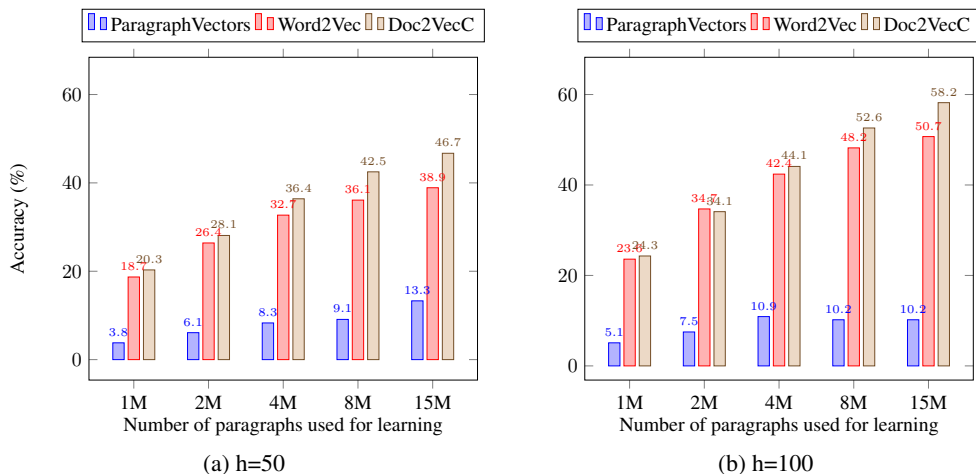

Figure 2: Accuracy on subset of the Semantic-Syntactic Word Relationship test set. Only questions containing words from the most frequent 30k words are included in the test.

| Semantic questions | Word2Vec | Doc2VecC | Syntactic questions | Word2Vec | Doc2VecC |
|---|---|---|---|---|---|
| capital-common-countries | 73.59 | **81.82** | gram1-adjective-to-adverb | 19.25 | **20.32** |
| capital-world | 67.94 | **77.96** | gram2-opposite | 14.07 | **25.54** |
| currency | 17.14 | 12.86 | gram3-comparative | 60.21 | **74.47** |
| city-in-state | 34.49 | **42.86** | gram4-superlative | 52.87 | **55.40** |
| family | 68.71 | 64.62 | gram5-present-participle | 56.34 | **65.81** |
| | | | gram6-nationality-adjective | 88.71 | **91.03** |
| | | | gram7-past-tense | 47.05 | **51.86** |
| | | | gram8-plural | 50.28 | **61.27** |
| | | | gram9-plural-verbs | 25.38 | **39.69** |

Table 4: Top 1 accuracy on the 5 type of semantics and 9 types of syntactic questions.

We trained the word embeddings of different methods using the English news dataset released under the ACL workshop on statistical machine translation. The training set includes close to 15M paragraphs with 355M tokens. We compare the performance of word embeddings trained by different methods with increasing embedding dimensionality as well as increasing training data.

We observe similar trends as in Mikolov et al. (2013a). Increasing embedding dimensionality as well as training data size improves performance of the word embeddings on this task. However, the improvement is diminishing. Doc2VecC produces word embeddings which performs significantly better than the ones generated by Word2Vec. We observe close to 20% uplift when we train on the full training corpus. Paragraph vectors on the other hand performs surprisingly bad on this dataset. Our hypothesis is that due to the large capacity of the model architecture, Paragraph Vectors relies mostly on the unique document vectors to capture the information in a text document instead of learning the word semantic or syntactic similarities. This also explains why the PV-DBOW Le & Mikolov (2014) model architecture proposed in the original work, which completely removes word embedding layers, performs comparable to the distributed memory version.

In table 5, we list a detailed comparison of the performance of word embeddings generated by Word2Vec and Doc2VecC on the 14 subtasks, when trained on the full dataset with embedding of size 100. We can see that Doc2VecC significantly outperforms the word embeddings produced by Word2Vec across almost all the subtasks.

## 4.4 DOCUMENT CLASSIFICATION

For the document classification task, we use a subset of the wikipedia dump, which contains over 300,000 wikipedia pages in 100 categories. The 100 categories includes categories under sports,

Table 5: Classification error (%) of a linear classifier trained on various document representations on the Wikipedia dataset.

| Model | BOW | DEA | Word2Vec + AVG | Word2Vec + IDF | ParagraphVectors | Doc2VecC |
|---|---|---|---|---|---|---|
| $h = 100$ | 36.03 | 32.30 | 33.2 | 33.16 | 35.78 | **31.92** |
| $h = 200$ | 36.03 | 31.36 | 32.46 | 32.48 | 34.92 | **30.84** |
| $h = 500$ | 36.03 | 31.10 | 32.02 | 32.13 | 33.93 | **30.43** |
| $h = 1000$ | 36.03 | 31.13 | 31.78 | 32.06 | 33.02 | **30.24** |

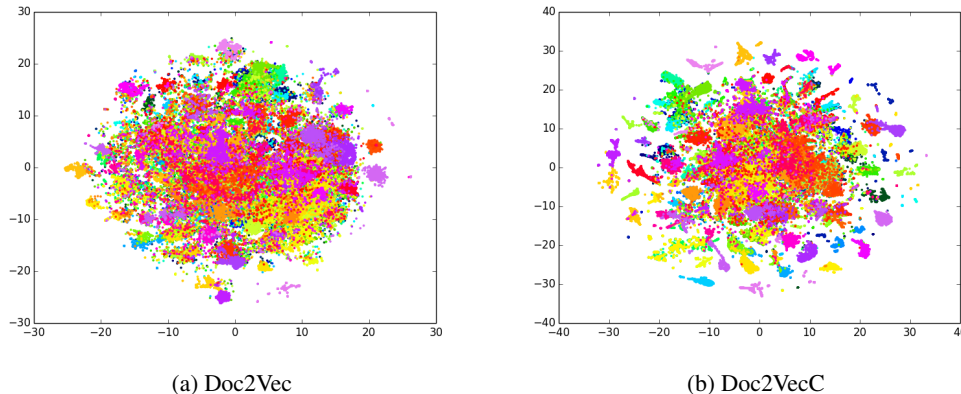

(a) Doc2Vec  (b) Doc2VecC

Figure 3: Visualization of document vectors on Wikipedia dataset using t-SNE.

entertainment, literature, and politics etc. Examples of categories include American drama films, Directorial debut films, Major League Baseball pitchers and Sydney Swans players. Body texts (the second paragraph) were extracted for each page as a document. For each category, we select 1,000 documents with unique category label, and 100 documents were used for training and 900 documents for testing. The remaining documents are used as unlabeled data. The 100 classes are balanced in the training and testing sets. For this data set, we learn the word embedding and document representation for all the algorithms using all the available data. We apply a cutoff of 10, resulting in a vocabulary of size $107,691$.

Table 5 summarizes the classification error of a linear SVM trained on representations of different sizes. We can see that most of the algorithms are not sensitive to the size of the vector representation. Doc2Vec benefits most from increasing representation size. Across all sizes of representations, Doc2VecC outperform the existing algorithms by a significant margin. In fact, Doc2VecC can achieve same or better performance with a much smaller representation vector.

Figure 3 visualizes the document representations learned by Doc2Vec (left) and Doc2VecC (right) using t-SNE (Maaten & Hinton, 2008). We can see that documents from the same category are nicely clustered using the representation generated by Doc2VecC. Doc2Vec, on the other hand, does not produce a clear separation between different categories, which explains its worse performance reported in Table 5.

Figure 4 visualizes the vector representation generated by Doc2VecC w.r.t. coarser categorization. we manually grouped the 100 categories into 7 coarse categories, television, albums, writers, musicians, athletes, species and actors. Categories that do no belong to any of these 7 groups are not included in the figure.

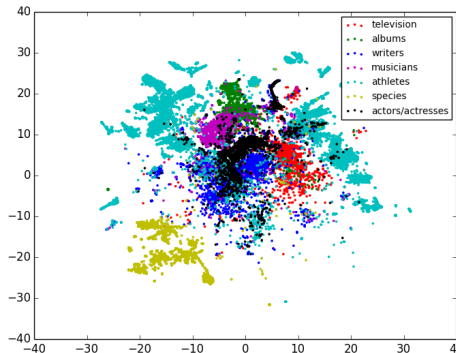

Figure 4: Visualization of Wikipedia Doc2VecC vectors using t-SNE.

We can see that documents belonging to a coarser category are grouped together. This subset includes is a wide range of sports descriptions, ranging from football, crickets, baseball, and cycling etc., which explains why the athletes category are less concentrated. In the projection, we can see documents belonging to the musician category are closer to those belonging to albums category than those of athletes or species.

## 4.5    SEMANTIC RELATEDNESS

We test Doc2VecC on the SemEval 2014 Task 1: semantic relatedness SICK dataset (Marelli et al., 2014). Given two sentences, the task is to determine how closely they are semantically related. The set contains 9,927 pairs of sentences with human annotated relatedness score, ranging from 1 to 5. A score of 1 indicates that the two sentences are not related, while 5 indicates high relatedness. The set is splitted into a training set of 4,500 instances, a validation set of 500, and a test set of 4,927.

We compare Doc2VecC with several winning solutions of the competition as well as several more recent techniques reported on this dataset, including bi-directional LSTM and Tree-LSTM[3] trained from scratch on this dataset, Skip-thought vectors learned a large book corpus [4] (Zhu et al., 2015) and produced sentence embeddings of 4,800 dimensions on this dataset. We follow the same protocol as in skip-thought vectors, and train Doc2VecC on the larger book corpus dataset. Contrary to the vocabulary expansion technique used in (Kiros et al., 2015) to handle out-of-vocabulary words, we extend the vocabulary of the learned model directly on the target dataset in the following way: we use the pre-trained word embedding as an initialization, and fine-tune the word and sentence representation on the SICK dataset. Notice that the fine-tuning is done for sentence representation learning only, and we did not use the relatedness score in the learning. This step brings small improvement to the performance of our algorithm. Given the sentence embeddings, we used the exact same training and testing protocol as in (Kiros et al., 2015) to score each pair of sentences: with two sentence embedding $\mathbf{u}_1$ and $\mathbf{u}_2$, we concatenate their component-wise product, $\mathbf{u}_1 \cdot \mathbf{u}_2$ and their absolute difference, $|\mathbf{u}_1 - \mathbf{u}_2|$ as the feature representation.

Table 6 summarizes the performance of various algorithms on this dataset. Despite its simplicity, Doc2VecC significantly out-performs the winning solutions of the competition, which are heavily feature engineered toward this dataset and several baseline methods, noticeably the dependency-tree RNNs introduced in (Socher et al., 2014), which relies on expensive dependency parsers to compose sentence vectors from word embeddings. The performance of Doc2VecC is slightly worse than the LSTM based methods or skip-thought vectors on this dataset, while it significantly outperforms skip-thought vectors on the IMDB movie review dataset (11.70% error rate vs 17.42%). As we hypothesized in previous section, while Doc2VecC is better at handling longer paragraphs, LSTM-based methods are superior for relatively short sentences (of length in the order of 10s). We would like to point out that Doc2VecC is much faster to train and test comparing to skip-thought vectors. It takes less than 2 hours to learn the embeddings on the large book corpus for Doc2VecC on a desktop with Intel i7 2.2Ghz cpu, in comparison to the 2 weeks on GPU required by skip-thought vectors.

## 5    CONCLUSION

We introduce a new model architecture Doc2VecC for document representation learning. It is very efficient to train and test thanks to its simple model architecture. Doc2VecC intrinsically makes sure document representation generated by averaging word embeddings capture semantics of document during learning. It also introduces a data-dependent regularization which favors informative or rare words while dampening the embeddings of common and non-discriminative words. As such, each document can be efficiently represented as a simple average of the learned word embeddings. In comparison to several existing document representation learning algorithms, Doc2VecC outperforms not only in testing efficiency, but also in the expressiveness of the generated representations.

---

[3]The word representation was initialized using publicly available 300-dimensional Glove vectors trained on 840 billion tokens of Common Crawl data

[4]The dataset contains 11,038 books with over one billion words

Table 6: Test set results on the SICK semantic relatedness task. The first group of results are from the submission to the 2014 SemEval competition; the second group includes several baseline methods reported in (Tai et al., 2015); the third group are methods based on LSTM reported in (Tai et al., 2015) as well as the skip-thought vectors (Kiros et al., 2015).

| Method | Pearson's $\gamma$ | Spearman's $\rho$ | MSE |
|---|---|---|---|
| Illinois-LH | 0.7993 | 0.7538 | 0.3692 |
| UNAL-NLP | 0.8070 | 0.7489 | 0.3550 |
| Meaning Factory | 0.8268 | 0.7721 | 0.3224 |
| ECNU | 0.8279 | 0.7689 | 0.3250 |
| Mean vectors (Word2Vec + avg) | 0.7577 | 0.6738 | 0.4557 |
| DT-RNN (Socher et al., 2014) | 0.7923 | 0.7319 | 0.3822 |
| SDT-RNN (Socher et al., 2014) | 0.7900 | 0.7304 | 0.3848 |
| LSTM (Tai et al., 2015) | 0.8528 | 0.7911 | 0.2831 |
| Bidirectional LSTM (Tai et al., 2015) | 0.8567 | 0.7966 | 0.2736 |
| Dependency Tree-LSTM (Tai et al., 2015) | 0.8676 | 0.8083 | 0.2532 |
| combine-skip (Kiros et al., 2015) | 0.8584 | 0.7916 | 0.2687 |
| Doc2VecC | 0.8381 | 0.7621 | 0.3053 |

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
