# Peer review of "Efficient Vector Representation for Documents through Corruption"

_ICLR 2017 — accepted_

[Public Comment · (anonymous) · 16 Dec 2016]
**Need more recent baselines**

Unsupervised document representations is an active area of research, so it would be useful to benchmark against something more recent than doc2vec, which was in ICML 2014. Skip-thought vectors, in particular, should really be included.

[Official Review · AnonReviewer2 · rating 7 · confidence 4 · 16 Dec 2016]
**Interesting corruption mechanism for document representation**
soundness 5 · clarity 4 · recommendation (unofficial) 5

This paper proposes learning document embeddings as a sum of the constituent word embeddings, which are jointly learned and randomly dropped out ('corrupted') during training. While none of the pieces of this model are particularly novel, the result is an efficient learning algorithm for document representation with good empirical performance.

Joint training of word and document embeddings is not a new idea, nor is the idea of enforcing the document to be represented by the sum of its word embeddings (see, e.g. '“The Sum of Its Parts”: Joint Learning of Word and Phrase Representations with Autoencoders' by Lebret and Collobert). Furthermore, the corruption mechanism is nothing other than traditional dropout on the input layer. Coupled with the word2vec-style loss and training methods, this paper offers little on the novelty front.

On the other hand, it is very efficient at generation time, requiring only an average of the word embeddings rather than a complicated inference step as in Doc2Vec. Moreover, by construction, the embedding captures salient global information about the document -- it captures specifically that information that aids in local-context prediction. For such a simple model, the performance on sentiment analysis and document classification is quite encouraging.

Overall, despite the lack of novelty, the simplicity, efficiency, and performance of this model make it worthy of wider readership and study, and I recommend acceptance.

[Official Review · AnonReviewer3 · rating 7 · confidence 3 · 19 Dec 2016]
**Simple idea, nicely composed**
soundness 4 · clarity 5

This paper discusses a method for computing vector representations for documents by using a skip-gram style learning mechanism with an added regularizer in the form of a global context vector with various bits of drop out. While none of the individual components proposed in this paper are new, I believe that the combination in this fashion is. Further, I appreciated the detailed analysis of model behaviour in section 3.

The main downside to this submission is in its relative weakness on the empirical front. Arguably there are more interesting tasks than sentiment analysis and k-way classification! Likewise, why waste 2/3 of a page on t-sne projections rather than use that space for further analysis?

While I am a bit disappointed by this reduced evaluation and agree with the other reviewers concerning soft baselines, I think this paper should be accepted: it's an interesting algorithm, nicely composed and very efficient, so it's reasonable to assume that other readers might have use for some of the ideas presented here.

[Official Review · AnonReviewer1 · rating 6 · confidence 4 · 19 Dec 2016 (modified: 04 Jan 2017)]
clarity 5 · recommendation (unofficial) 5

This paper presents a framework for creating document representations. 
The main idea is to represent a document as an average of its word embeddings with a data-dependent regularization that favors informative or rare words while forcing common words to be close to 0. 
Experiments on sentiment analysis and document classification show that the proposed method has the lowest error rates compared to baseline document embedding methods. 

While I like the motivation of finding the best way to encode a document into a vector, the paper does not offer significant technical contributions.
Most of the techniques are not new, and the main selling point is the simplicity and speed of the proposed method. 
For this reason, I would like to see good results for more than two tasks to be convinced that this is the best way to learn document representations.  
For RNN-LM, is the LM trained to minimize classification error, or is it trained  as a language model? Did you use the final hidden state as the representation, or the average of all hidden states?
One of the most widely used method to represent documents now is to have a bidirectional LSTM and concatenate the final hidden states as the document representation. 
I think it would be useful to know how the proposed method compares to this approach for tasks such as document classification or sentiment analysis.

[Author Response · Minmin Chen · 21 Dec 2016]
**revisions**

Dear reviewers, 

Thank you for your feedback. The updated manuscript included skip-thought as another baseline method. We will test this idea on more datasets, in particular the ones experimented in Skip-thought vectors in the submission.

[Author Response · Minmin Chen · 24 Dec 2016]
**new dataset and baselines added**

Dear reviewers, 

I added another dataset to the draft (appendix): SemEval 2014 Task 1 semantic relatedness SICK dataset, as several of the recent works the reviewers pointed out have been reported on this dataset. Despite its simplicity, Doc2VecC significantly out-performs the winning solutions of the competition and several baseline methods, noticeably the dependency-tree RNNs introduced in [1], which relies on additional dependency parsers to compose sentence vectors from word embeddings. The performance of Doc2VecC is comparable (slightly worse) than the LSTM based methods or skip-thought vectors on this dataset. On the other hand, it is much faster to train and test. As reported in the original paper, training of the skip-thought vector models on the book corpus dataset takes around 2 weeks on GPU. In contrast, it takes less than 2 hours to learn the embeddings for Doc2VecC on a desktop with Intel i7 2.2Ghz cpu.

I also provided more insight on why Skip-thought vectors did not perform well on the movie review dataset in Section 4.2 (Accuracy). 

I would greatly appreciate it if you could take another look at revisions and provide me with some feedbacks. 

[1] Socher, Richard, et al. "Grounded compositional semantics for finding and describing images with sentences." Transactions of the Association for Computational Linguistics 2 (2014): 207-218.

[Public Comment · (anonymous) · 22 Mar 2017]
**About Table 3: Words with embeddings closest to 0 learned by different algorithms.**

I'm using

[Author Response · Minmin Chen · 18 May 2017]
**Better word embeddings**

Extended the paper with experiments on the word relationship dataset, showing Doc2VecC generates better word embeddings in comparison to Word2Vec or Paragraph Vectors.

[Final Decision · Program Chairs · 06 Feb 2017]
**ICLR committee final decision**

The introduced method for producing document representations is simple, efficient and potentially quite useful. Though we could quibble a bit that the idea is just a combination of known techniques, the reviews generally agree that the idea is interesting.
 
 Pros:
 + interesting and simple algorithm
 + strong performance
 + efficient
 
 Cons:
 + individual ideas are not so novel
 
 This is a paper that will be well received at a poster presentation.